# Health Status Recognition Method for Rotating Machinery Based on Multi-Scale Hybrid Features and Improved Convolutional Neural Networks

**DOI:** 10.3390/s23125688

**Published:** 2023-06-18

**Authors:** Xiangang Cao, Xingyu Guo, Yong Duan, Fuqiang Zhang, Hongwei Fan, Xin Xu

**Affiliations:** 1School of Mechanical Engineering, Xi’an University of Science and Technology, Xi’an 710054, China; caoxg@xust.edu.cn (X.C.); duanyong152@163.com (Y.D.); fuqiangzhang3010@163.com (F.Z.); hw_fan@xust.edu.cn (H.F.); xuxin@xust.edu.cn (X.X.); 2Shaanxi Province Key Laboratory of Intelligent Detection and Control of Mining Electromechanical Equipment, Xi’an 710054, China

**Keywords:** rotating machinery, multi-scale hybrid features, convolutional neural networks, health status identification

## Abstract

Rotating machinery is susceptible to harsh environmental interference, and fault signal features are challenging to extract, leading to difficulties in health status recognition. This paper proposes multi-scale hybrid features and improved convolutional neural networks (MSCCNN) health status identification methods for rotating machinery. Firstly, the rotating machinery vibration signal is decomposed into intrinsic modal components (IMF) using empirical wavelet decomposition, and multi-scale hybrid feature sets are constructed by simultaneously extracting time-domain, frequency-domain and time-frequency-domain features based on the original vibration signal and the intrinsic modal components it decomposes. Secondly, using correlation coefficients to select features sensitive to degradation, construct rotating machinery health indicators based on kernel principal component analysis and complete health state classification. Finally, a convolutional neural network model (MSCCNN) incorporating multi-scale convolution and hybrid attention mechanism modules is developed for health state identification of rotating machinery, and an improved custom loss function is applied to improve the superiority and generalization ability of the model. The bearing degradation data set of Xi’an Jiaotong University is used to verify the effectiveness of the model. The recognition accuracy of the model is 98.22%, which is 5.83%, 3.30%, 2.29%, 1.52%, and 4.31% higher than that of SVM, CNN, CNN + CBAM, MSCNN, and MSCCNN + conventional features, respectively. The PHM2012 challenge dataset is used to increase the number of samples to validate the model effectiveness, and the model recognition accuracy is 97.67%, which is 5.63%, 1.88%, 1.36%, 1.49%, and 3.69% higher compared to SVM, CNN, CNN + CBAM, MSCNN, and MSCCNN + conventional features methods, respectively. The MSCCNN model recognition accuracy is 98.67% when validated on the degraded dataset of the reducer platform.

## 1. Introduction

With the development of industrialization, rotating machinery has achieved a wide range of applications among industries [1]. Affected by the harsh environmental interference and non-smooth working conditions during operation [2,3], the critical components of rotating machinery are prone to deformation, cracks and fractures, and other damages, and if they fail, they will cause incalculable economic losses to average production and even threaten personal safety. Therefore, the study of rotating machinery health status identification method is of great significance to improve the operational reliability and stability of rotating machinery [4].

The vibration signal of rotating machinery monitoring contains a lot of information, and the equipment health status identification method based on vibration signal has become a research hot spot. Rotating machinery health status recognition is mainly divided into methods based on signal processing, methods based on the combination of feature extraction and pattern recognition, and methods based on deep learning, among which the construction of feature engineering is the crucial step of rotating machinery health status recognition. The quality of the feature set directly determines the accuracy of health status recognition. Currently, related scholars have studied the extraction of statistical parameters in the time domain, frequency domain, and time-frequency domain based on signal processing methods, and have achieved specific results [5,6]. Liu [7] used the generalized S-transform to transform the original vibration signal in the time-frequency domain to obtain a two-dimensional time-frequency matrix and the singular value decomposition of the eigenmatrix to obtain the particular value vector group characterizing the characteristic information of the specific state parts of cylindrical roller bearings to achieve the identification of different state types of cylindrical roller bearings. Huang [8] used Transformer to extract features to obtain fine-grained better feature representation, combined with adversarial domain training and local maximum mean difference to guide feature learning by minimizing the difference in distribution between source and target domains resulting in better diagnosis. Choudhary [9] uses multi-input convolutional neural network technology to integrate the features of vibration signals and sound signals, then convert them into time-frequency spectrum, thus improving the accuracy of fault diagnosis. However, when coupled components are present in mechanical devices and harsh and complex environments, the devices tend to exhibit nonlinear behavior. Therefore, simple statistical feature parameter extraction cannot accurately identify fault characteristics. In the current research, few studies have unified advanced signal processing methods by extracting statistical parameters or entropy indicators in the time, frequency, and time-frequency domains [10], to construct complete multiscale hybrid domain feature sets to represent the degradation trend of rotating machinery.

In recent years, experts and scholars at home and abroad in the research of equipment health status identification methods can be broadly divided into three categories: knowledge-driven, model-driven, and data-driven [11]. In order to avoid problems such as a large amount of human experience required for knowledge-driven and the difficulty of establishing mathematical degradation models for model-driven, the data-driven health status identification methods have gained widespread attention [12]. Traditional data-driven methods for equipment health status identification based on data mainly include Bayesian classification [13], support vector machines [14], and artificial neural networks [15]. With the rapid development of deep learning, deep learning-related models have been developed in the field of rotating machinery health status identification due to their nonlinear solid mapping capability [16,17]. Lei [18] used mechanical frequency domain signals to train deep neural networks, combined unsupervised and supervised learning, and adaptively extracted extensive data fault features to identify the health conditions of multi-stage gear trains effectively. Chen [19] proposed a deep convolutional neural network-based health state identification method for planetary gearboxes, fusing the raw data of horizontal and vertical vibration signals and using a deep network to extract features for identification automatically. Zhang [20] stacked multiple self-encoders and classification layers to construct an integrated model of state feature self-learning and state recognition to achieve state recognition of rolling bearings. Chen [21] proposed a rolling bearing life state identification method based on source domain multi-sample integration, using random sampling to obtain multiple training samples within the source domain by multiple equal random selections of inter-class pieces in the source domain, constructing a kernel classifier using the kernel matrix and outputting the state identification results. However, the above methods still have shortcomings, such as difficulty in feature extraction and low accuracy of model recognition.

To address the above problems, the health status recognition method for rotating machinery based on multi-scale hybrid features and improved convolutional neural networks is proposed. The main contributions of this paper are summarized as follows:Empirical wavelet decomposition (EWT) of vibration signals, extraction of time domain, frequency domain and time-frequency-domain features on the decomposed signals, and establishment of multi-scale hybrid features sets.The correlation coefficients are used to select features sensitive to degradation. Then the kernel principal component analysis method (KPCA) is used to construct health state identification indexes and complete the state category classification.The convolutional neural network model is improved from the perspectives of multi-scale features extraction, attention mechanism to screen optimal feature information, and optimization of losses to improve the accuracy of the health status of crucial components of rotating machinery.

## 2. Basic Principles

### 2.1. Empirical Wavelet Transform

The empirical wavelet decomposition (EWT) constructs a set of filters by adaptive partitioning of the Fourier spectrum to obtain empirical wavelet functions with orthogonal properties, and performs empirical wavelet transform on the partitioned interval to propose amplitude-modulated frequency modulated single component components with tightly supported Fourier spectrum, which is better adaptive than wavelet transform and empirical modal decomposition, and does not have the problems of modal confusion and sizeable computational effort. EWT [22,23,24] essentially decomposes the signal into a sum of *N* + 1 eigenmode functions, which is calculated as follows:(1)$$X\left(t\right)=\sum_{i=0}^{N}X_{i}(t)$$
where:
$X\left(t\right)$
is the time domain signal;
$X_{i}(t)$
is the useful amplitude modulation frequency component.

Firstly, segment the signal spectrum $X\left(w\right)$, build a wavelet filter bank, normalize the signal Fourier spectrum to [0,π], and then decompose it into N continuous spaces, use $w_{n}$ to represent the boundary of each continuous space, each interval is defined as ${⋀}_{n}=[w_{n-1},w_{n}]$, obviously there are $\overset{N}{\underset{n=1}{⋃}}\wedge_{n}=[0,\pi ]$, and then take $w_{n}$ as the center frequency and $2\tau_{n}$ as the transition segment of adjacent intervals. For the determined N intervals, the empirical wavelet is defined as a bandpass filter on each interval, and Gilles constructs the empirical wavelet according to the construction method of Meyer wavelet, and uses the empirical wavelet function and the empirical scale function to decompose the signal by EWT. The empirical scale function and empirical wavelet function are calculated as:(2)$$\phi_{n}\left(w\right)=\left\{\begin{array}{c} 1\\ \cos\left[\frac{\pi }{2}\beta \left(\frac{1}{2\tau_{n}}\left(\left|w\right|-w_{n}+\tau_{n}\right)\right)\right]\\ \sin\left[\frac{\pi }{2}\beta \left(\frac{1}{2\tau_{n}}\left(\left|w\right|-w_{n}+\tau_{n}\right)\right)\right]\\ 0 \end{array}\right.$$
(3)$$\psi_{n}\left(w\right)=\left\{\begin{array}{c} 1\\ \cos\left[\frac{\pi }{2}\beta \left(\frac{1}{2\tau_{n}}\left(\left|w\right|-w_{n}+\tau_{n}\right)\right)\right]\\ \sin\left[\frac{\pi }{2}\beta \left(\frac{1}{2\tau_{n}}\left(\left|w\right|-w_{n}+\tau_{n}\right)\right)\right]\\ 0 \end{array}\right.$$
where:
β
is an arbitrary
$C^{k}\left(\left[0,1\right]\right)$
function that satisfies:
(4)$$\beta \left(x\right)=\left\{\begin{array}{c} 0,\;\;x\leq 0\\ 1,\;\;x\geq 1 \end{array}\right.$$

The signal reconstruction is decomposed by EWT to obtain the IMF component $f_{i}(t)$ defined as:(5)$$f_{0}\left(t\right)=W_{\int }^{\varepsilon }\left(0,t\right)*\emptyset_{1}(t)$$
(6)$$f_{i}\left(t\right)=W_{\int }^{\varepsilon }\left(i,t\right)*\psi_{n}(t)$$

### 2.2. Convolutional Neural Network

There are various structures of convolutional neural networks [25] (CNN). Still, the basic framework is similar and generally consists of an input layer, a convolutional layer, a pooling layer, a fully connected layer, and an output layer, with the different layers connected by means of feature mapping [26].

The fundamentals of each layer are:

The input layer is responsible for initially feeding the raw data into the neural network. The input layer is characterized as M-dimensional.

The convolutional layer is the core layer of CNN, which extracts the features of the input samples and outputs them to the next layer through the convolutional kernel. Convolution has two new features compared to the traditional fully connected neural network: one is the sparse connection, which connects with some neurons in the previous layer, and this area of the connected neuron is also called the perceptual field; the other is weight sharing, where each convolution kernel completes all processing with only one set of weight parameters. The convolutional operation is formulated as:(7)$$X_{j}^{n}=f\left(\sum_{i=1}^{M}X_{i}^{n-1}\times W_{ij}^{n}+b_{j}\right)$$
where: $X_{j}^{n}$ denotes the jth feature mapping of the nth layer, f denotes the activation function, M denotes the number of input feature mappings, $X_{i}^{n-1}$ denotes the ith feature mapping of the $\left(n-1\right)$th layer, $W_{ij}^{n}$ denotes the weights, and $b_{j}$ denotes the bias.

The pooling layer is a way to reduce the dimensionality of the input data, thus representing the signal with high-quality features and reducing the amount of data processing in the next layer, preventing the overfitting of the model. The standard pooling methods are maximum, mean, random, and median pooling. In this paper, the maximum pooling is chosen, and the calculation formula is:(8)$$y^{n}=max\left(a_{i}^{n}\right),\;\;\;i=1,2,\ldots ,k$$
where:
n
denotes the number of feature layers, and
$a_{i}^{n}$
denotes the *i*th feature of the *n*th layer.

The fully-connected layer is to connect the pooled features to convert them into 1×N dimensional matrices corresponding to each of the N outcomes to achieve classification. Each neuron in the fully connected layer is connected to all neurons in the previous layer. Therefore, the training parameters in this layer account for a larger proportion.

The output layer is then connected to the fully connected layer and outputs the final classification result. The structure of the convolutional neural network is shown in Figure 1.

## 3. Methodology of this Paper

The flow of the rotating machinery health state assessment model based on multi-scale hybrid features and improved convolutional neural network (MSCCNN) established in this paper is shown in Figure 2. The specific contents of this paper are as follows: Firstly, multi-scale hybrid feature extraction of the original vibration signal of rotating machinery. Secondly, feature dimensionality reduction and health indicator construction for rotating machinery. Finally, the MSCCNN model for health status identification of rotating machinery.

### 3.1. Multi-Scale Hybrid Features Extraction

When a device fails, the relevant signal characteristics change simultaneously, including the amplitude and probability distribution in the time domain, the energy at different frequencies in the frequency domain, the position of the leading edge of the primary energy spectrum, and the structural distribution of the time-frequency power. Feature extraction is the key to health status identification. Single-scale features can identify and analyze the equipment under specific working conditions, but it isn’t easy to reflect the operation status of the machinery comprehensively. In complex working conditions, the decomposition of vibration signals and the establishment of hybrid features at multiple scales can characterize the health state of rotating machinery more completely. For the above problems, EWT has a good effect on the decomposition of vibration signals of rotating machinery. In order to fully express the degradation characteristics of rotating machinery, this paper proposes a multi-scale hybrid features extraction method, which combines EWT decomposition to decompose the original vibration signal of rotating machinery at a deeper level, and extracts multi-scale hybrid features in the time domain, frequency domain, and time-frequency domain for the original vibration signal and its intrinsic modal components.

In this paper, the multiscale hybrid features are extracted simultaneously in the time domain, frequency domain, and time-frequency domain for the rotating machine vibration signal and the third-order IMF components of the vibration signal decomposition. The vibration signals of the rotating machine are extracted in X and Y directions, and the original vibration signals in X direction and their third-order IMF components are extracted with 30 features in the time domain, frequency domain, and time-frequency domain, respectively, which means that 120 features are extracted from the four vibration signals. One hundred twenty features are extracted from the original vibration signals in Y direction by the same operation, which means that 240 features are extracted in total.

A total of 12 features are extracted in the time domain, among which the dimensioned features include mean, variance, standard deviation, maximum value, minimum value, peak-to-peak value, root mean square value, absolute mean value, and square root amplitude, the dimensionless features include waveform indicator, peak indicator, and pulse indicator, whose expressions are shown in Table 1. For the convenience of experimental statistics, the time domain features extracted from the original vibration signal in the X direction are called XV-F_1_ to XV-F_12_, and the time domain features extracted from its first order IMF components are named XIMF1-F_1_ to XIMF1-F_12_. The naming of the time domain features extracted from the second and third order IMF components is the same as that of the time domain features extracted from the first order IMF components.

A total of 10 features are extracted from the frequency domain, including spectral mean value, spectral mean square heel value, frequency domain frequency skewness, frequency domain frequency kurtosis, the center of gravity frequency, root mean square frequency, frequency amplitude variance, and frequency domain frequency kurtosis index, which reflect the magnitude of vibration energy in the frequency domain. Frequency domain analysis of vibration signals first requires the conversion of the time domain waveform of the movement into spectral information with the help of the discrete Fourier transform, as follows:(9)$$S\left(k\right)=\sum_{k=0}^{N-1}x\left(k∆tz\right)e^{\frac{-2\pi jnk}{N}},(n=1,2,\ldots ,N-1)$$
where: $x\left(k∆tz\right)$ is the sampled value of the vibration signal; N is the number of sampling points; ∆t is the sampling interval; k is the ordinal number of the discrete value in the time domain.

After obtaining the spectral information, the frequency domain index calculation formula is shown in Table 2. For the convenience of experimental statistics, the frequency domain features extracted from the original vibration signal in the X direction are named XV-F_13_ to XV-F_22_, the frequency domain features extracted from its first order IMF components are named XIMF1-F_13_ to XIMF1-F_22_. The naming of the frequency domain features extracted from the second and third order IMF components is the same as that of the frequency domain features extracted from the first order IMF components.

In order to facilitate the experimental statistics, the time-frequency domain features extracted from the original vibration signal in X direction are named XV-F_23_ to XV-F_30_, the time-frequency domain features extracted from its first order IMF components are named XIMF1-F_23_ to XIMF1-F_30_. The naming of the time-frequency domain features extracted from the second and third order IMF components is the same as that of the time-frequency domain features extracted from the first order IMF components. The calculation formula is:(10)$$E_{l}=E_{l}/\sum_{i=1}^{n}E_{i},l\epsilon [1,n]$$
where: $E_{l}$ is the wavelet energy spectrum of the original signal on n scales. When the scale is a, the wavelet energy spectrum is calculated as follows:(11)$$E\left(a\right)=\int_{-\infty }^{\infty }{\left|W_{f}(a,b)\right|}^{2}db$$
where:
$W_{f}(a,b)$
is the amplitude of the wavelet transform.

### 3.2. Multi-Scale Features Selection and Health Index Construction

To select the most representative degradation features, eliminate redundant information, ensure that the selected features have good degradation performance and improve the classification accuracy, this paper uses correlation coefficients [27] to complete the selection of degradation-sensitive features. In order to better satisfy the health state classification, a good performance health degradation index needs to be established, and the KPCA algorithm [28] is selected to fuse the optimal multiscale hybrid features after selection, and the health index is constructed according to the cumulative contribution rate of the principal elements. The specific steps are:

First, the data are nonlinearly mapped from space R into the high-dimensional space H:(12)ϕ:R→H

The covariance matrix of the sample is expressed as:(13)$$C^{H}=\frac{1}{Z}\sum_{i=1}^{Z}\phi (x_{i}){\phi (x_{i})}^{T}$$
where:
$x_{i}(i=1,2,\ldots ,Z)$
is the sequence sample,
Z
is the input sequence length, and
$\phi (x_{i})$
is the feature sample point.

Solving the characteristic equation for $C^{H}$:(14)$$C^{H}V=\lambda_{q}V$$
where: V is the eigenvector and $\lambda_{q}$ is the eigenvalue. Multiplying $\phi (x_{i})$ equally on both sides of the above equation:(15)$$\phi \left(x_{i}\right)C^{H}V=\lambda_{q}\phi \left(x_{i}\right)V$$

After refining
V, the above equation is expressed as:
(16)$$\frac{1}{Z}\sum_{\gamma =1}^{Z}\alpha_{\gamma }\left\{\sum_{\omega =1}^{Z}\left[\phi \left(x_{i}\right)\phi \left(x_{\omega }\right)\phi \left(x_{\omega }\right)\phi \left(x_{\gamma }\right)\right]\right\}=\lambda_{q}\sum_{\gamma =1}^{Z}\left[\phi \left(x_{i}\right)\phi \left(x_{\gamma }\right)\right]$$
where:
γ
and
ω
are different spatial signal samples, and
$\alpha_{\gamma }$
is the presence coefficient.

To avoid solving the unspecifiable representation of $\phi \left(x_{i}\right)$, the kernel matrix $K_{\gamma \omega }=\left[\phi \left(x_{\gamma }\right)\phi \left(x_{\omega }\right)\right]$ is introduced and the above equation is simplified as:(17)$$Z\lambda_{q}\alpha_{\mu }=K_{\gamma \omega }\alpha_{\mu }$$
where: $\alpha_{\mu }={\left(\alpha_{1},\alpha_{2},\ldots ,\alpha_{n}\right)}^{T}$.

Finally, feature downscaling fusion is performed, the feature values are arranged in descending order, η is the cumulative contribution rate threshold, and the cumulative contribution rate of the first J features in the total number of Q features is:(18)$$\sum_{q=1}^{J}\lambda_{q}/\sum_{q=1}^{Q}\lambda_{q}\geq \eta$$

According to expert experience, in order to achieve good results in KPCA screening of degraded features, a threshold value η≥85% is generally set to determine the J most important principal features. This paper uses the first principal feature as a health indicator to complete the health state.

### 3.3. Rotating Machinery Health Identification (MSCCNN) Modeling

The structure of the rotating machinery health recognition (MSCCNN) model is shown in Figure 3. Firstly, a multi-scale convolution layer is built to extract multi-scale features under the different field of view sizes. Secondly, the mixed features after multi-scale convolution are spliced and fused, convolved, and pooled to obtain higher level features. The features are input to the attention mechanism module (CBAM) to complete adaptive feature extraction, and the extracted features are convolved and pooled again to extract more abstract features. Finally, the output of the classification results is performed through a fully connected layer, and the model parameters are optimized by improving the loss.

(1)Multi-scale convolution

The multi-scale convolution uses convolution kernels of different sizes to convolve the input features to obtain features at different convolution scales, then splices the features at different convolution scales to receive the fused multi-scale features, and finally integrates the features through pooling. The multi-scale convolution can enrich the proposed data features, extract sensitive feature information from the input data from a global perspective, and then improve the segmentation performance of the data. The multi-scale convolution is shown in Figure 4.

(2)Attention mechanism module

The attentional mechanism (CBAM) is a simple and effective attentional module for feed-forward convolutional neural networks. Given an intermediate feature map, the CBAM module sequentially infers attention along two independent dimensions, channel and space, and subsequently multiplies them by the input features to perform adaptive feature extraction. Because CBAM is a lightweight, general-purpose module, it can be seamlessly integrated into any convolutional neural network architecture with negligible consumption and can be trained end-to-end with the underlying convolutional neural network. The CBAM structure is shown in Figure 5.

(3)Custom loss function

The loss function is a non-negative real-valued function that measures the degree of inconsistency between the predicted and actual values of the model. The smaller the value of its loss function, the better the robustness of the model. Convolutional neural networks use an error backpropagation algorithm to train the model. The MSCCNN model in this paper uses the “MAE,” “MSE,” and “Log cosh” loss functions and a modified custom loss function (LM) to perform the test. The “LM” loss function combines the advantages of the “Log cosh” and “MSE” loss functions and avoids their respective disadvantages. The improved custom loss function (LM) formula in this paper is:(19)$$MSE=\sum_{i=1}^{n}{\left(f\left(x\right)-y\right)}^{2}/n$$
(20)$$L\left(y,f\left(x\right)\right)=\sum_{i=1}^{n}log(cosh\left(y-f\left(x\right)\right))$$
(21)LM=0.8L+0.2MSE
where
y
denotes the true value and
$f\left(x\right)$
denotes the predicted value.

The innovation of the MSCCNN model built in this paper:

(1) Enhancement of feature extraction field of view and enrichment of feature information by multi-scale convolution. (2) Introducing a self-attention mechanism to complete degradation-sensitive feature self-extraction. (3) Customized loss function to ensure optimization of model parameters and improve model robustness.

## 4. Experimental Verification and Analysis

### 4.1. Validation of Bearing Degradation Data Set at Xi’an Jiaotong University

#### 4.1.1. XJTU-SY Data Set Introduction

In this paper, the effectiveness of the proposed method is verified using accelerated life test data of XJTU-SY bearing, and the experimental bench consists of an AC motor, speed controller, rotating shaft, support bearing, and test bearing [29]. The experimental data consisted of failure data of 15 bearings in total for three different operating conditions. The accelerated life experiment platform of rolling bearings of Xi’an Jiaotong University is shown in Figure 6, and the bearing data information is shown in Table 3.

In this paper, the experimental data of bearings 1, 2, 3, and 4 under working condition one and bearings 1, 3, 4, and 5 under working condition two are selected as the training set, with a total of 1969 sample data. The sample data of the training set is divided into two parts: validation and test. The ratio of validation samples to test samples is 8:2.

#### 4.1.2. XJTU-SY Data Set Experimental Verification and Analysis

(1) Multi-scale features extraction and selection.

Taking bearing 1 under working condition one as an example, a multi-scale hybrid features set is extracted and constructed. First, its data set is decomposed to third-order using EWT. Then the original vibration signal and the third order IMF components are extracted from the time domain, frequency domain, and time-frequency domain with a total of 30 features. Since the test bearing has two kinds of original vibration signals, the horizontal original vibration signal and its third-order IMF components are extracted with a total of 120 features, and the vertical original vibration signal and its third-order IMF components are extracted with a total of 120 features, so a total of 240 multi-scale hybrid features are extracted after processing. The processing results are shown in Table 4.

Secondly, the correlation coefficient is used to select features in the multi-scale mixed features vector set that can better characterize the fault degradation process of bearing 1 under operating condition one. In this paper, 60 multi-scale mixed features with high ratings are selected to form the optimal feature vector set. The multi-scale hybrid features selection and the comparison of superior difference features are shown in Figure 7.

Finally, the processed multi-scale mixed features vector set is then fused and downscaled by KPCA to output the health status index of the bearing. The constructed rolling bearing health state index is shown in Figure 8.

According to the constructed health status index, analysis of domestic and international literature related to the field of research equipment health identification reveals that equipment performance degradation can usually be classified into four states [30,31]. Therefore, combined with the fundamental operation of rolling bearings and expert experience, the rolling bearing health status is divided into four levels [32], and each level and the corresponding health status index interval are shown in Table 5.

(2) MSCCNN model training and validation.

The MSCCNN model and training parameters built in this paper are as follows: ① The 60 features are convolved by convolution of 3 channels and 3 scales, with a convolution kernel size of 3, 5, and 7, channel size of 32, and activation function of Relu activation function. ② The multi-scale hybrid features are stitched by the Add function and input to the pooling layer for pooling with a pooling step of 2. ③ The convolution operation is performed by 1 convolution layer with a convolution kernel size of 3 and a channel size of 64, and using the Relu activation function. ④ The features are input to the CBAM module for more advanced feature extraction, followed by convolutional operation of the features through one convolutional layer, and the convolved features are input to the pooling layer, where the convolutional kernel size is 3, the channel size is 32, the pooling step is 2, and the activation function is the Relu activation function. ⑤ The features after the convolution are input to the last convolutional layer for the convolution operation, where the convolution kernel size is 5, the channel size is 16, the pooling step size is 2, and the activation function is the Relu activation function. ⑥ Finally, the bearing health status is classified into four categories: health, good, deterioration and failure through the full connection layer and “Softmax” function. ⑦ The optimizer is Adam, the learning rate is 0.001, and the training times are 200.

During the experiment, the optimizer and loss function of the model have a significant impact on how well the model is trained. When the model is trained, firstly, the improved custom loss function is selected, and the optimizer is selected from SGD, RMSprop, and Adam for comparison. Through the experiment, the model reaches the optimum, and the model accuracy is the highest when the Adam optimizer is selected. The recognition accuracy results of different optimizers are shown in Figure 9.

Then, after the optimizer of the model is determined, the loss functions “MAE,” “MSE,” and “Log cosh” are selected for comparison with the improved custom loss functions. Through experiments, the enhanced custom loss function (LM) is chosen to make the highest accuracy of the model in this paper. It is proved that the improved custom loss function in this paper outperforms the “MAE,” “MSE,” and “Log cosh” loss functions. The recognition accuracies of different loss functions are shown in Figure 10.

In summary, the finalized optimizer of the MSCCNN model is Adam, and the improved custom loss function is selected as the loss function of the model in this paper. When the parameters of the MSCCNN model are optimal, the confusion matrix of the test results of the model for bearing health status identification in the validation sample is shown in Figure 11.

The accuracy of the model in assessing the health status of the bearings as measured by the test samples is as high as 98.22%, and the accuracy of identification is good in each state.

To further validate the accuracy and superiority of the MSCCNN health state assessment model proposed in this paper, it is compared with support vector machine (SVM), CNN, CNN + CBAM, MSCNN, and MSCCNN + conventional features (60 features in the time domain, frequency domain and time-frequency domain) [33], respectively, and the same training set is used to conduct the experiments. The confusion matrix results of each method for the accuracy of bearing health status identification are shown in Figure 12, and the specific data are shown in Table 6.

From the specific data in Figure 12 and Table 6, it can be seen that the multi-scale hybrid features and improved convolutional neural network (MSCCNN) proposed in this paper are more accurate for the bearing health status identification method, with 5.83%, 3.30%, 2.29%, 1.52%, and 4.31% improvement compared to SVM, CNN, CNN + CBAM, MSCNN, and MSCCNN + conventional features methods, respectively.

#### 4.1.3. XJTU-SY Data Set MSCCNN Model Generalization Performance Validation

Data samples of Bearing1_5, Bearing2_2, and Bearing3_3 are selected for testing to verify the model generalization capability. The results of the MSCCNN model for bearing health status identification are shown in Table 7.

From Table 7, the accuracy of the MSCCNN model for health status identification of three bearings is 88.46%, 87.58%, and 98.11%, respectively, with an average accuracy of 91.38%, which proves that the MSCCNN model has a good generalization ability for health status identification of bearings.

### 4.2. PHM2012 Challenge Data Set Validation

#### 4.2.1. PHM2012 Data Set Introduction

The data set [34] was provided by the PRONOSTIA experimental platform of the FEMTO-ST Institute, which describes the degradation of ball bearings throughout their service life and consists of three main parts, including the rotating part, the degradation generating part (load part) and the measuring part. The raw vibration signal is collected by installing acceleration sensors on the outer ring of the test stand, and the radial load force is applied to accelerate the degradation of the ball bearing. The bearing vibration signals are divided into two directions, horizontal and vertical, with a sampling rate of 25.6 kHz, sampled every 10 s and lasting 0.1 s, so 2560 data points are collected each time. 3 operating conditions and 17 bearings are included in the PHM2012 challenge data set. The specific information of the bearing operating conditions data used in this paper is shown in Table 8.

#### 4.2.2. PHM2012 Data Set MSCCNN Experimental Results and Analysis

In this paper, the degraded data of bearing 1 and 3 under working condition one, bearing 1 under working condition two, and bearing 2 under working condition three are selected to form the training set with a total of 7726 sample data. The sample data of the training set is divided into two parts: validation and test, and the ratio of validation samples to test samples is 8:2.

By the method of this paper, the health status of the ball bearing is identified, and the sample classification and identification results are shown in Table 9.

As shown in Table 9, the multi-scale hybrid features and improved convolutional neural network (MSCCNN) proposed in this paper are more accurate in identifying the health status of ball bearings, with 5.63%, 1.88%, 1.36%, 1.49%, and 3.69% improvement compared to the SVM, CNN, CNN + CBAM, MSCNN, and MSCCNN + traditional features, respectively.

To verify the generalization ability of the model, the degradation data of bearing 2 under working condition two and bearing 3 under working condition three were selected for testing. The accuracy of the MSCCNN model for identifying the health status of the two bearings was 80.93% and 84.10%, respectively, with an average accuracy of 82.52%, which proved that the MSCCNN model has good generalization ability for identifying the health status of the bearings.

### 4.3. Reducer Platform Degradation Data Set Validation

#### 4.3.1. Platform Introduction

A rotating machinery (reducer) experimental platform is built to verify the effectiveness of the proposed method. The laboratory platform mainly consists of a magnetic powder brake, a gear reducer, an AC motor, a shaft coupling, and various sensors. Vibration, speed, temperature, and other sensors are installed to collect signals during the operation of the reducer. The specific types of sensors and their installation locations are shown in Figure 13.

The data acquisition card model of the decelerator experimental platform is MCC USB-1608FS-Plus, the vibration acceleration sensor model is CT1050LC with a sampling frequency of 5 kHz, and the speed and torque sensor model is HCNJ-101 with a sampling frequency of 1 kHz. The vibration sensor CT1050LC is installed in the horizontal and vertical direction of the input shaft of the reducer. The frequency response range is 0.5–1.5 kHZ, and the vibration measurement range is 0–10 g. The torque sensor measures the torque range of 0–200 N.m, the torque output is 5–15 KHz, and the measuring speed is 0–3000 r/min. The sensor sends the collected signals to the cloud server for storage and then carries out relevant data analysis by reading the data stored in the server database.

Since vibration signals contain rich state information, the following two parameters are selected in this paper as the data basis for the health state identification of the reducer. The specific parameters are shown in Table 10. In the experiment, the vibration data sampling frequency is 5000 Hz, and the vibration signal is recorded once in 1 min.

#### 4.3.2. Decelerator Data Experimental Results and Analysis

The multi-scale optimal and worst features curves extracted in this paper are shown in Figure 14.

From the figure, it can be seen that the good characteristic curves in the multi-scale hybrid features have obvious monotonicity, which can well characterize the degradation status of the reducer with time. The health state index of the reducer constructed by multi-scale hybrid features selection and KPCA dimensionality reduction is shown in Figure 15.

The health status identification of the reducer by the method of this paper, the sample classification and the identification results confusion matrix are shown in Figure 16, and the specific data is shown in Table 11.

From the specific data in Figure 16 and Table 11, it can be seen that the multi-scale hybrid features and improved convolutional neural network (MSCCNN) proposed in this paper are more accurate in identifying the health status of the reducer platform, with 6.95%, 4.30%, 2.64%, 1.98%, and 5.62% improvement compared to SVM, CNN, CNN + CBAM, MSCNN, and MSCCNN + conventional features methods respectively.

## 5. Conclusions

This paper addresses the problem that rotating machinery is susceptible to severe environmental disturbances and challenging to extract fault signal features leading to difficult health status identification. A health status recognition method for rotating machinery based on multi-scale hybrid features and improved convolutional neural networks is proposed to improve the accuracy of health status recognition of rotating machinery. The multi-scale hybrid features are extracted based on the modal components of the empirical wavelet decomposition. The correlation coefficient is used to select the degradation-sensitive features, and the kernel principal component analysis is used to construct the rotating machinery health indexes based on which the health state classification is completed. A convolutional neural network model incorporating multi-scale convolution and hybrid attention mechanism modules is developed for the health status identification of rotating machines, and the loss function is customized to improve the superiority and generalization ability of the model. The accuracy of model identification was 98.22% on the bearing degradation dataset of Xi’an Jiaotong University and 98.67% on the gearbox platform degradation dataset. It effectively solves the problem of difficulty in identifying the health status of rotating machinery due to the difficulty in extracting data features under severe working conditions.

Future research should: (1) study signal noise reduction methods under strong noise background and signal feature extraction methods sensitive to degradation; (2) study signal feature extraction and selection methods under variable working conditions and establish rotating machinery health state identification models under complex working conditions. (3) deep learning evaluation models are poorly interpretable, and subsequent research should be conducted on model interpretability and feature visualization.

## Figures and Tables

**Figure 1 sensors-23-05688-f001:**
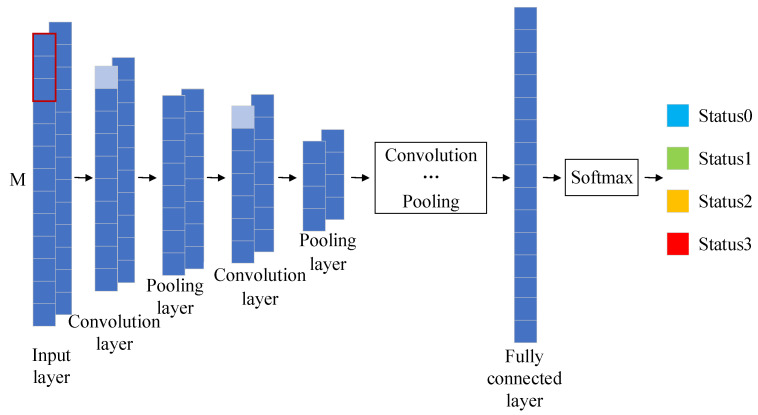
Convolutional neural network structure diagram.

**Figure 2 sensors-23-05688-f002:**
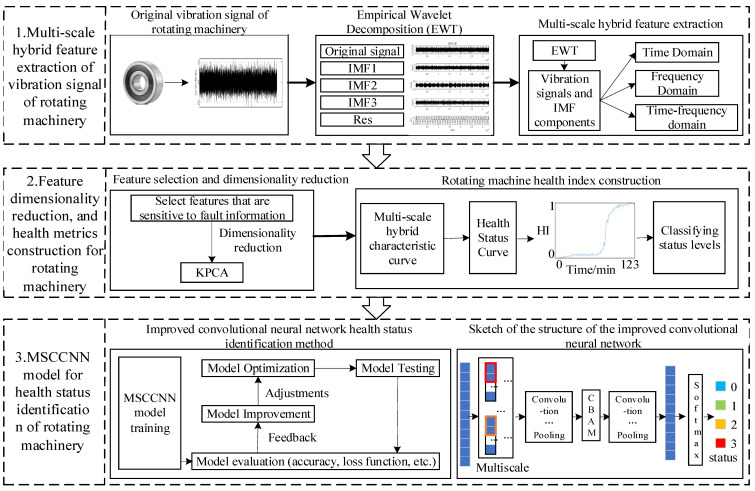
MSCCNN flow chart.

**Figure 3 sensors-23-05688-f003:**
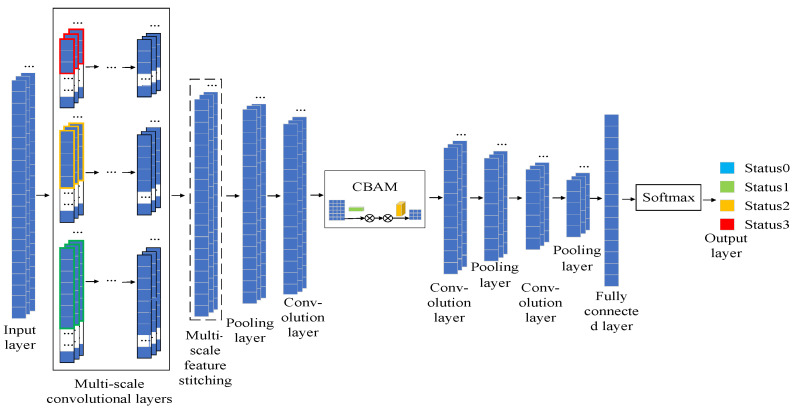
MSCCNN model.

**Figure 4 sensors-23-05688-f004:**
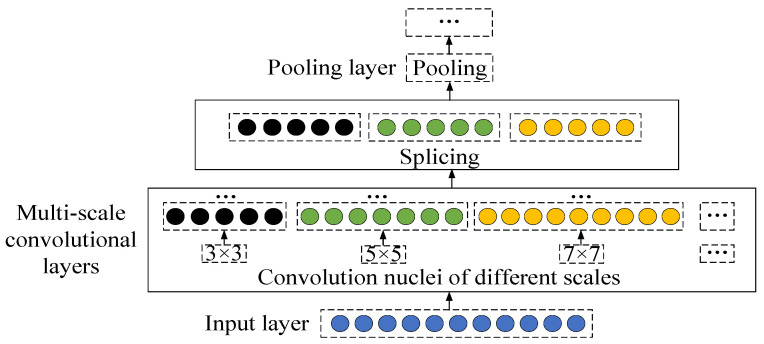
Multi-scale convolutional structure.

**Figure 5 sensors-23-05688-f005:**
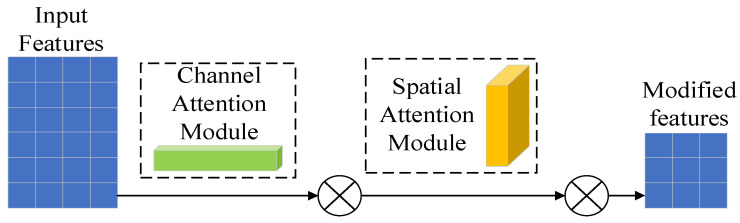
CBAM structure.

**Figure 6 sensors-23-05688-f006:**
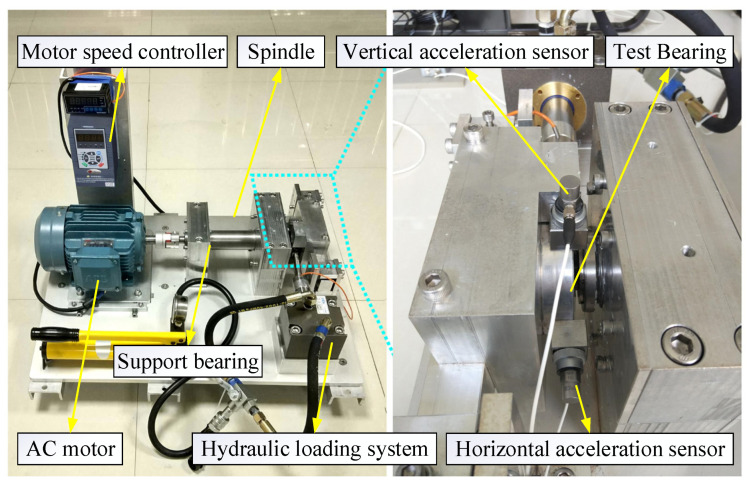
XJTU-SY experimental platform.

**Figure 7 sensors-23-05688-f007:**
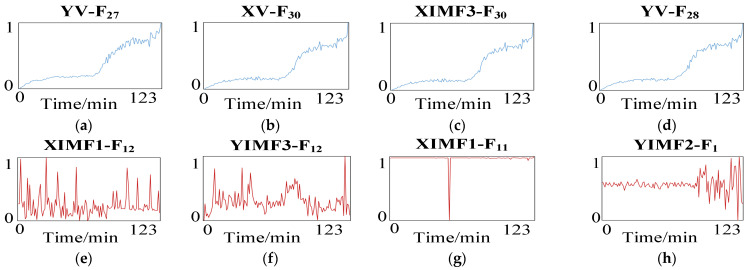
Multi-scale hybrid features selection. (**a**) The first feature; (**b**) The second feature; (**c**) The third feature; (**d**) The fourth feature; (**e**) The 237th feature; (**f**) The 238th feature; (**g**) The 239th feature; (**h**) The 240th feature.

**Figure 8 sensors-23-05688-f008:**
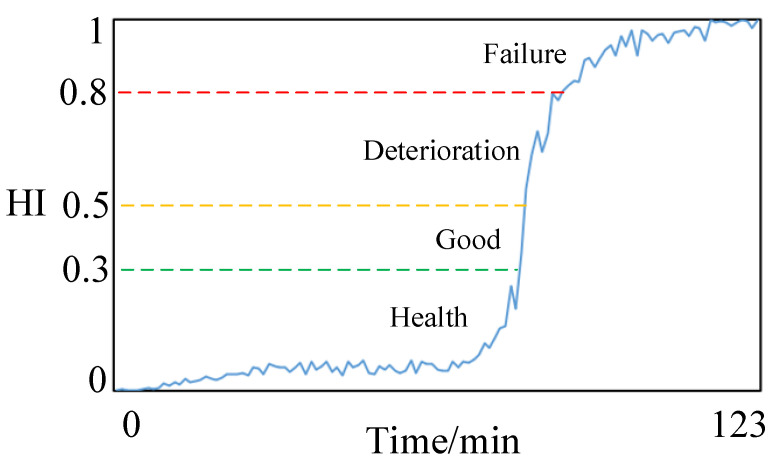
The health status index of rolling bearing is constructed.

**Figure 9 sensors-23-05688-f009:**
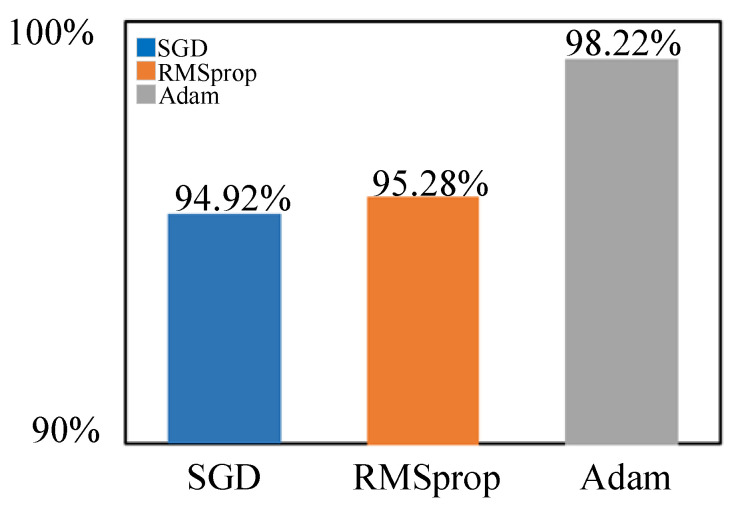
Recognition accuracy of different optimizers.

**Figure 10 sensors-23-05688-f010:**
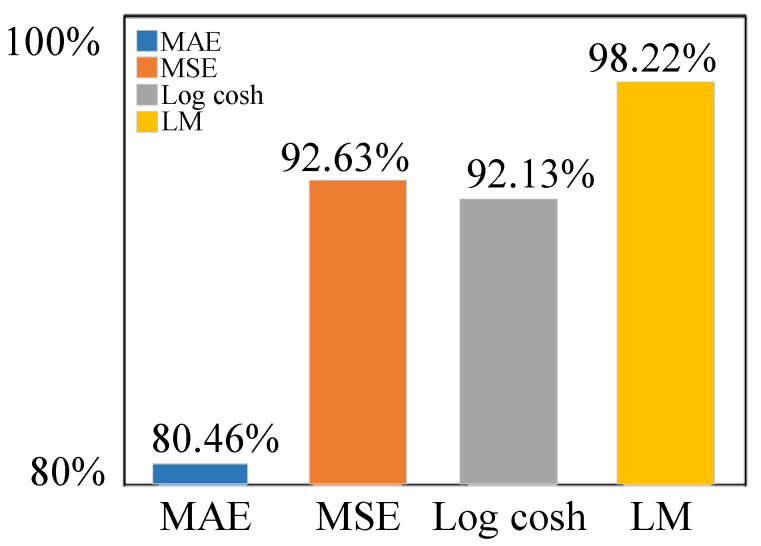
Recognition accuracy of different loss function.

**Figure 11 sensors-23-05688-f011:**
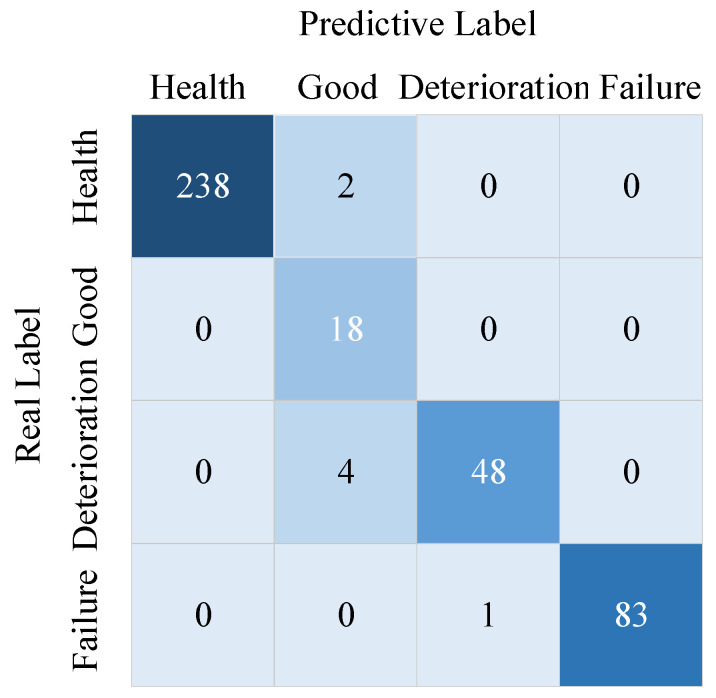
The test set identifies the result confusion matrix.

**Figure 12 sensors-23-05688-f012:**
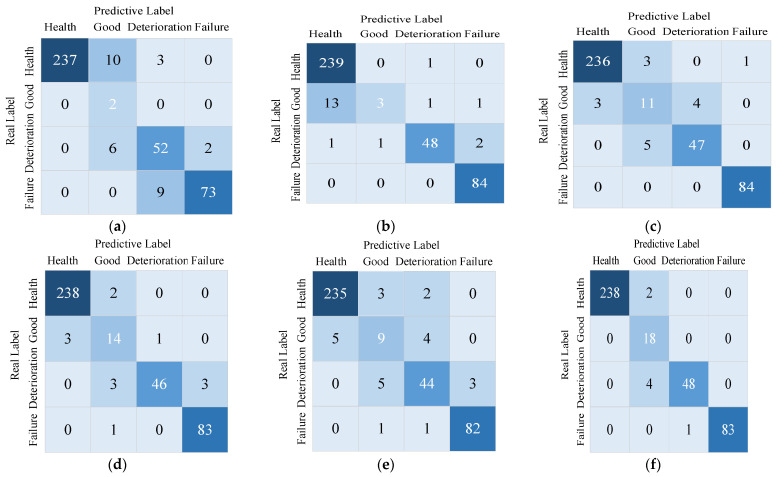
Identification method confusion matrix results in XJTU-SY data set. (**a**) SVM; (**b**) CNN; (**c**) CNN + CBAM; (**d**) MSCNN; (**e**) MSCCNN + conventional features; (**f**) MSCCNN.

**Figure 13 sensors-23-05688-f013:**
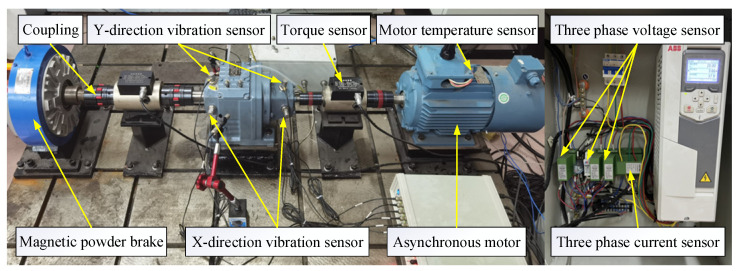
Reducer experimental platform.

**Figure 14 sensors-23-05688-f014:**
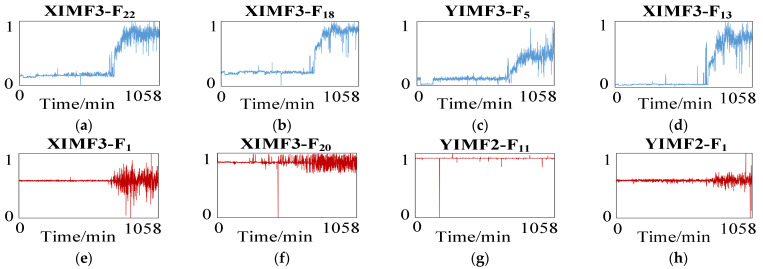
Selection of reducer degradation characteristics. (**a**) The first feature; (**b**) The second feature; (**c**) The third feature; (**d**) The fourth feature; (**e**) The 237th feature; (**f**) The 238th feature; (**g**) The 239th feature; (**h**) The 240th feature.

**Figure 15 sensors-23-05688-f015:**
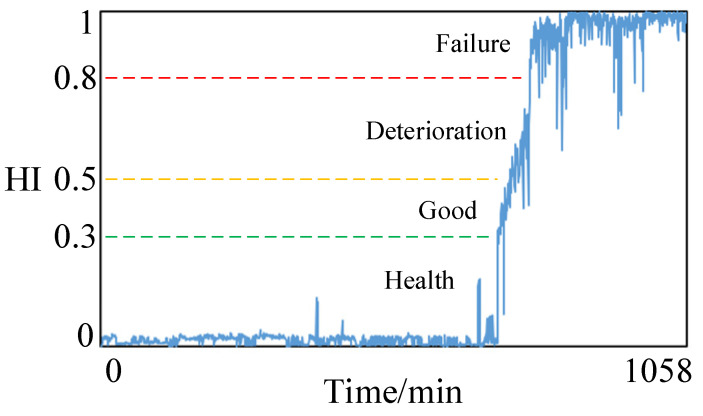
The health status index of reducer is constructed.

**Figure 16 sensors-23-05688-f016:**
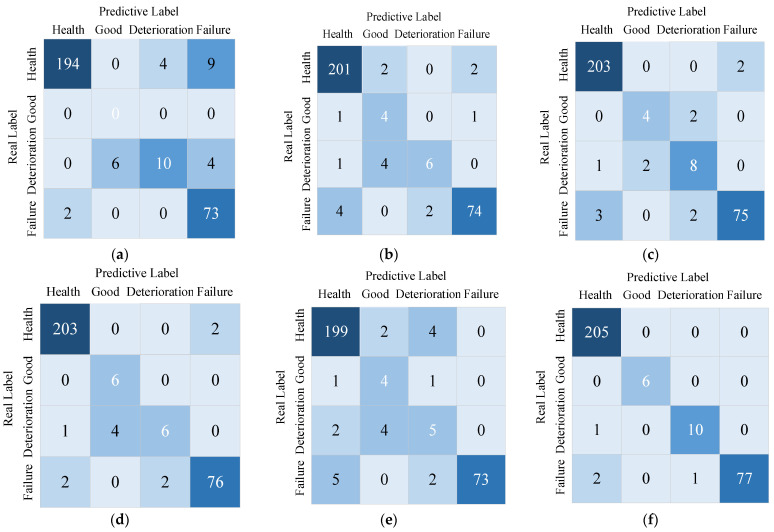
Identification method confusion matrix results in reducer data set. (**a**) SVM; (**b**) CNN; (**c**) CNN + CBAM; (**d**) MSCNN; (**e**) MSCCNN + conventional features; (**f**) MSCCNN.

**Table 1 sensors-23-05688-t001:** Time domain characteristic parameter expression.

Parameter Expressions	Parameter Expressions
$F_{1}=\frac{1}{N}\sum_{i=1}^{N}x_{i}$	$F_{7}=\sqrt{\frac{1}{N}\sum_{i=1}^{N}x_{i}^{2}}$
$F_{2}=\frac{1}{N}\sum_{i=1}^{N}x_{i}^{2}$	$F_{8}=max\left\{\left|x_{i}\right|\right\}$
$F_{3}=\frac{1}{N}\sum_{i=1}^{N}{\left(x_{i}-F_{1}\right)}^{2}$	$F_{9}=|max\left(x_{i}\right)-min\left(x_{i}\right)|$
$F_{4}=\sqrt{\frac{1}{N-1}\sum_{i=1}^{N}{\left(x_{i}-F_{1}\right)}^{2}}$	$F_{10}=\frac{F_{7}}{F_{1}}$
$F_{5}={\left(\frac{1}{N}\sum_{i=1}^{N}\sqrt{\left|x_{i}\right|}\right)}^{2}$	$F_{11}=\frac{F_{8}}{F_{7}}$
$F_{6}=\frac{1}{N}\sum_{i=1}^{N}\left|x_{i}\right|$	$F_{12}=\frac{F_{8}}{F_{1}}$

**Table 2 sensors-23-05688-t002:** Frequency domain characteristic parameter expression.

Parameter Expressions	Parameter Expressions
$F_{13}=\frac{\sum_{k=1}^{K}S\left(k\right)}{K}$	$F_{18}=\sqrt{\frac{\sum_{k=1}^{K}{\left(f_{k}-F_{17}\right)}^{2}S\left(k\right)}{K-1}}$
$F_{14}=\sqrt{\frac{\sum_{k=1}^{K}{\left(S\left(k\right)-F_{13}\right)}^{2}}{K-1}}$	$F_{19}=\sqrt{\frac{\sum_{k=1}^{K}{f_{k}}^{2}S\left(k\right)}{\sum_{k=1}^{K}S\left(k\right)}}$
$F_{15}=\frac{\sum_{k=1}^{K}{\left(S\left(k\right)-F_{13}\right)}^{3}}{\left(K-1\right){F_{14}}^{3}}$	$F_{20}=\sqrt{\frac{\sum_{k=1}^{K}{f_{k}}^{4}S\left(k\right)}{\sum_{k=1}^{K}{f_{k}}^{2}S\left(k\right)}}$
$F_{16}=\frac{\sum_{k=1}^{K}{\left(S\left(k\right)-F_{13}\right)}^{4}}{\left(K-1\right){F_{14}}^{4}}$	$F_{21}=\frac{\sum_{k=1}^{K}{f_{k}}^{2}S\left(k\right)}{\sqrt{\sum_{k=1}^{K}S\left(k\right)\sum_{k=1}^{K}{f_{k}}^{4}S\left(k\right)}}$
$F_{17}=\frac{\sum_{k=1}^{K}f_{k}S\left(k\right)}{\sum_{k=1}^{K}S\left(k\right)}$	$F_{22}=\frac{F_{18}}{F_{17}}$

**Table 3 sensors-23-05688-t003:** Bearing data set information table.

Work Conditions	Data Set	Total Sample Size	Data Set	Total Sample Size
1.35 Hz12 KN	Bearing1_1	123	Bearing1_4	122
	Bearing1_2	161	Bearing1_5	52
	Bearing1_3	158		
2.37.5 Hz11 KN	Bearing2_1	491	Bearing2_4	42
	Bearing2_2	161	Bearing2_5	339
	Bearing2_3	533		
3.40 Hz10 KN	Bearing3_1	2538	Bearing3_4	1515
	Bearing3_2	2496	Bearing3_5	114
	Bearing3_3	371		

**Table 4 sensors-23-05688-t004:** Multi-scale mixed features score.

Serial Number	Features	Score	Serial Number	Features	Score
1	YV-F_27_	0.912114076	……	……	……
2	XV-F_30_	0.838685397	231	YV-F_1_	0.048443585
3	XIMF3-F_30_	0.837374829	232	YIMF1-F_1_	0.048391187
4	YV-F_28_	0.833849017	233	XIMF2-F_11_	0.045665439
5	YIMF3-F_27_	0.833573272	234	YIMF2-F_11_	0.040269179
6	YV-F_28_	0.823099802	235	XIMF2-F_1_	0.039946343
7	YIMF3-F_30_	0.813190966	236	XV-F_11_	0.032247291
8	YIMF3-F_28_	0.797538005	237	XIMF1-F_12_	0.026462805
9	XV-F_18_	0.796106301	238	YIMF3-F_12_	0.021388632
10	XV-F_28_	0.789489338	239	XIMF1-F_11_	0.020182963
……	……	……	240	YIMF2-F_1_	0.016714913

**Table 5 sensors-23-05688-t005:** Health status grade of rolling bearing.

Health Levels	Operation Status	Health Indicator Intervals	Grade Label
Health	Normal operation, no need for maintenance	0.0≤x<0.3	0
Good	Stable operation, planned maintenance	0.3≤x<0.5	1
Deterioration	Deterioration of operating condition, timely maintenance	0.5≤x<0.8	2
Failure	Failure to operate properly, repair	0.8≤x≤1.0	3

**Table 6 sensors-23-05688-t006:** Comparison of identification method results in XJTU-SY data set.

Method	Test Results (Number of Correct|Total)								Accuracy
	Health		Good		Deterioration		Failure		
SVM	237	250	2	2	52	60	73	82	92.39%
CNN	239	240	3	18	48	52	84	84	94.92%
CNN + CBAM	236	240	11	18	47	52	84	84	95.93%
MSCNN	238	240	14	18	46	52	83	84	96.70%
MSCCNN + conventional features	235	240	9	18	44	52	82	84	93.91%
MSCCNN	238	240	18	18	48	52	83	84	98.22%

**Table 7 sensors-23-05688-t007:** MSCCNN model recognition results.

Different Samples	Test Results (Number of Correct|Total)								Accuracy
	Health		Good		Deterioration		Failure		
Bearing1_5	38	39	0	1	1	2	7	10	88.46%
Bearing2_2	66	73	6	10	0	9	69	69	87.58%
Bearing3_3	338	344	0	1	6	6	20	20	98.11%

**Table 8 sensors-23-05688-t008:** PHM2012 Challenge data set.

Work Conditions	Data Set	Total Sample Size
1.4000 N 1800 r/min	Bearing1_1	2803
	Bearing1_3	2375
2.4200 N 1650 r/min	Bearing2_1	911
	Bearing2_2	797
3.5000 N 1500 r/min	Bearing3_2	1637
	Bearing3_3	434

**Table 9 sensors-23-05688-t009:** Comparison of results of recognition methods.

Method	Test Results (Number of Correct|Total)								Accuracy
	Health		Good		Deterioration		Failure		
SVM	475	494	148	195	446	471	354	386	92.04%
CNN	493	507	164	174	464	496	360	369	95.79%
CNN + CBAM	496	507	166	174	458	496	369	369	96.31%
MSCNN	489	507	165	174	473	496	360	369	96.18%
MSCCNN + conventional features	503	507	153	174	444	496	353	369	93.98%
MSCCNN	499	507	169	174	478	496	364	369	97.67%

**Table 10 sensors-23-05688-t010:** Health state monitoring parameters of reducer.

Serial Number	Parameter Name	Unit
1	X-directional vibration of reducer input shaft	g
2	Y-directional vibration of reducer input shaft	g

**Table 11 sensors-23-05688-t011:** Comparison of identification method results in reducer data set.

Method	Test Results (Number of Correct|Total)								Accuracy
	Health		Good		Deterioration		Failure		
SVM	194	207	0	0	10	20	73	75	91.72%
CNN	201	205	4	6	6	11	74	80	94.37%
CNN + CBAM	203	205	4	6	8	11	75	80	96.03%
MSCNN	203	205	6	6	6	11	76	80	96.39%
MSCCNN + conventional features	199	205	4	6	5	11	73	80	93.05%
MSCCNN	205	205	6	6	10	11	77	80	98.67%

## Data Availability

The data set of the reducer platform provided in this study is available from the corresponding author upon request. Due to project confidentiality, these data will not be made public.

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
