# Peer review of "Health Status Recognition Method for Rotating Machinery Based on Multi-Scale Hybrid Features and Improved Convolutional Neural Networks"

_sensors, 2023, doi:10.3390/s23125688_

Round 1

Reviewer 1 Report

This paper introduces a multi-scale hybrid features and improved convolutional neural networks (MSCCNN) health status identification method for rotating machinery. This study is a welcome contribution to the literature in the field of health status recognition. This study is interesting and the organization is well. However, some questions are required to address before accepted.

1 ) The health index in the paper may show different monotonicity and robustness for different data sets. It is recommended that the author use different data sets to verify the applicability of the rolling bearing health status evaluation criteria in Table 5. 

2 ) The health classification of rolling bearings is level 4 rather than level 3 or level 5, and the theoretical basis is what needs to be explained clearly. 

3 ) What is the innovation of MSCCNN method ?

The writing of the article is relatively standardized, and there are fewer grammatical errors. However, some sentences need to be improved.

1) This paper's multi-scale hybrid features are extracted simultaneously in the time domain, frequency domain, and time-frequency domain for the rotating machine vibration signal and its decomposed third-order IMF components. “This paper's multi-scale hybrid features……

2) The calculation formula is as follows:……

3) To verify the generalization ability of the model, the data samples of bearing 5 in condition one, bearing 2 in condition two, and bearing 3 in condition three were selected for testing.

4) The validity of the model is verified using the degraded dataset of bearings from Xi'an Jiaotong University, and the model recognition accuracy is 98.22%, which is improved by 5.83%, 3.30%, 2.29%, and 1.52% compared with SVM, CNN, CNN+CBAM, and MSCNN methods, respectively.

Reviewer 2 Report

1. The paper should be interesting.

2. Please provide a detailed explanation for the development principle of Health Indicator Intervals in Table 5, and indicate if it is universal.

3. The paper lacks the comparative experiments with other feature construction methods under the same network structure.

4. The paper should consider whether the size of the dataset is suitable for the designed model.

5. Some sentences in the article are too long and not easy for readers to read.

6. The sentence(Secondly, using correlation coefficients to select features sensitive to degradation, construct rotating machinery health indicators based on kernel principal component analysis and complete health state classification.)in the abstract lacks a subject.

Reviewer 3 Report

The manuscript approaches an interesting subject.

Some recommendations:

1. The applied training method used for the CNN has to be detailed in the manuscript. The corresponding equations has to be introduced in the manuscript. 

2. A study has be introduced in the manuscript regarding the possibility to use other types of neural networks in order to solve the approached technical problem.

3. All the graphs, even the small ones, have to have the quantities figured on the ordinates. 

Reviewer 4 Report

亲爱的作者们,

Below you will find a list of my comments on the article.

1. The authors did not directly provide information on the specific transducers used in their study, nor about their sensitivity, mass, or useful frequency range. The same applies to DAQ devices. For a scientific paper to be recognised, all information must be presented in order for the experiments to be reproducible to other scientists.

2. The authors presented the use of neural networks, particularly convolutional neural networks (CNNs), in their research. They described the basic elements of the network, such as the input layer, the convolutional layer, the pooling layer,, the fully connected layer and the output layer. The authors also described how these layers are connected by feature mapping. However, it is not clear that the authors have provided enough information about the process of training the neural network. For example, it is not clear what hyperparameters were used when training the network, such as the learning rate, the number of training epochs, whether a regularization mechanism such as dropout or L1/L2 regularization was used. Additionally, the authors did not provide information on the model validation process. It is unclear whether they used cross-validation, a validation set, or metrics to evaluate the model beyond accuracy.

3. The authors did not discuss potential limitations or challenges of using neural networks in their research. For example, neural networks can be prone to over-fitting if not properly regulated, and their results can be difficult to interpret due to their "black box" nature.

4. The issue of machine condition classification using neural networks is very popular in science today. Please clearly indicate why this method would be better than other methods? 

5. Both the research and data from Xi'an Jiaotong University are laboratory data from constructed test stands. The article in this topic is a lot. We only observe the state of the bearings without the influence of other sources. The method developed by the Authors should be verified on a test stand, but their validation should be based on data of existing machines from industry. Even if we introduce disturbances in the lab bench, it is still an active experiment. The key is the data from the passive or active-passive experiment. This is missing from the article.

6. In my opinion, the list of bibliography is insufficient. There is a lack of the description of achievements of other research teams from around the world. I give examples of research works for analysis:

https://doi.org/10.3390/sym11101212

https://doi.org/10.1016/j.promfg.2019.06.096

https://doi.org/10.5604/01.3001.0015.8254

https://doi.org/10.1007/s00170-018-2420-0

https://doi.org/10.1016/j.engappai.2022.105794

https://doi.org/10.1016/j.ifacol.2021.08.141

https://doi.org/10.1007/s13369-023-07664-5

https://doi.org/10.3233/JIFS-223012

https://doi.org/10.1007/s42417-022-00769-5

https://doi.org/10.1016/j.displa.2022.102233

https://doi.org/10.1016/j.engappai.2023.105872

亲切的问候,
Reviewer

Round 2

Reviewer 2 Report

The paper is now good enough to publish.